# Effects of Acoustic Environment Types on Stress Relief in Urban Parks

**DOI:** 10.3390/ijerph20021082

**Published:** 2023-01-07

**Authors:** Jun Zhang, Hongliang Yan, Dan Wang

**Affiliations:** School of Landscape Architecture, Northeast Forestry University, Harbin 150040, China

**Keywords:** urban park, acoustic environment, stress relief, electrocardiogram, heart rate variability

## Abstract

Urban public space environments are critical to the health of residents. In previous studies on urban park environments and health, landscape environment questionnaires have been the main method to evaluate the environmental quality and comfort of urban parks. The research on sound perception also focuses on the exploration of evaluation methods and evaluation indicators; there is little objective empirical evidence in these studies. To further explore the nature of the health role of urban parks, this study started with the sound types of urban parks, based on a field survey, combined the electrocardiogram (ECG) index with the sound type of the park through a portable intelligent device, and HR and RMSSD were selected as the ECG indicators to evaluate the stress relief status. The regression model between the type of acoustic environments and the ECG data was established through the analysis of relevant data. This paper tries to improve the physiological recovery benefit and influence mechanism of sound types in urban parks from an objective point of view and puts forward reasonable suggestions to improve the sound environment in urban parks. The preliminary results show that, in a short time frame, natural sound has a strong relieving effect on mental pressure, while mechanical sound has an obvious impediment effect on the recovery of mental pressure. The results also reveal that the human voice has no obvious impediment effect, and changes in wind and broadcast sound have little impact on the recovery of mental pressure.

## 1. Introduction

Since the 21st century, with the rapid development of society, the economy, and urbanization in China, the opportunities for citizens to be close to nature have been greatly reduced. High-intensity work, stressful lives, and changes in the living environment have indirectly affected the physical and mental health of urban residents [1,2,3]. The number of patients suffering from depression, diabetes, and cardiovascular or neurological diseases, which are mostly caused by mental stress, is increasing year by year [4]. Thus, mental stress has gradually attracted the attention of the public and researchers in various countries.

Compared with other urban spaces, urban parks have a higher number of green spaces and plant landscapes, which can reduce air pollution on a local scale [5], block some urban noise [6], and alleviate environment stresses such as the urban heat island effect [7], and these positive macro effects of urban parks on the urban environment enable them to provide residents with good outdoor activity spaces and access to the natural environment and promote the physical and mental health of residents [8]. Urban parks, as important players in the restorative environment of urban public space, have an important role in improving the urban ecological environment and enriching and relieving stress in citizens’ lives [9,10].

Attention recovery suggests that natural landscapes can trigger people’s natural attention and transfer people from the fatigue of directed attention, so that people’s attention can shift from directed attention fatigue, thus resting and replenishing their attention and cognitive abilities [11,12]. In addition, Ulrich’s [13,14] stress recovery theory also suggests that exposure to nature or urban greenery can provide mental stress relief and that this relief is an “immediate, subconscious stress response”. Research in recent years has also further confirmed the ability of natural spaces in cities to promote human health issues, as White [15] studied the degree of restorative ratings and the preferences of people for different landscape types through photographs of different landscape types. Wang [16] conducted physiological and psychological tests on Chinese university students aged 18–24 years through video scenes, and the results showed that different landscape features have different degrees of restorative qualities and that natural landscapes are more likely to reduce stress than hard landscapes [17]. Some of these studies have also delved into the effects of urban park landscape elements and various landscape components on psychophysiological activity [18]. These studies provide evidence for improving the visual quality and health restoration benefits of urban parks.

The quantification of only landscape environmental elements has led to the study of restorative environments in urban parks to focus mostly on visual perception, thus neglecting the restorative role of other senses, and the acoustic environment is also an important aspect in the study of restorative urban environments. Early studies on the link between sound and health found that various noises in the living environment may effect sleep [19], and long-term noise exposure can cause cardiovascular and cerebrovascular diseases and other adverse effects [20]. Some scholars have further refined the scope of the sound environment and restorative environment research and proposed the concept of soundscape, which is defined by ISO12913-1 as a sound environment perceived, experienced, or understood by a person or a group of people in a specific contextual situation [21]. Liu [22] used a questionnaire to examine people’s motivations for visiting urban green spaces and the relationship between visual landscape factors and the perceived occurrence and loudness of individual sounds, the preference for individual sounds, and the overall soundscape preference in urban green spaces. Li [23] evaluated the impact of audiovisual interactions on soundscape assessment and noise control. They summarized and discussed the methods of field investigations and laboratory experiments, thus constructing a classification system and conceptual framework for soundscape research and design. However, the previous studies mainly stayed at the subjective level, and the evidence of objective data analysis is still in the minority. Not only that, the physiology and neurophysiology of park sound environments are still in their infancy, and the mechanisms of action on users’ psychological and physiological responses are yet to be further studied.

Therefore, based on the research results, this study quantifies and analyzes the types of urban park sound environments and uses electrocardiography (ECG) as the physiological health evaluation index of mental stress to study the restorative effect of sound types on mental stress relief at the environmental and physiological levels to provide ideas for exploring the restorative research methods in park sound environments; to provide a new perspective for urban park environment research; and to provide a basis for improving the quality of urban park sound environments.

## 2. Materials and Methods

### 2.1. Investigation into Acoustic Environment Characteristics

Residential land can carry a more complete population hierarchy [24]. Therefore, this study chose an urban park with mainly residential land use as the study site. Xiangjiang Park in Harbin City was selected as the study site, which is located in the Nangang District of Harbin City and has a large number of trees in the park, and a sightseeing water system. The park site is surrounded by a variety of building properties such as residential communities, shopping malls, hotels, tourist attractions, office buildings, etc. Taking it as the research object of the acoustic environment study is closer to the daily life and activities of nearby residents than large municipal parks far from the city, and it is representative of studying the psychological and physiological mechanisms of pressure relief of urban residents in the park acoustic environment. The park location and surrounding environment are shown in Figure 1.

A field questionnaire was used to investigate the users’ evaluation of the park’s acoustic environment characteristics, from which we obtained the distribution of the park’s sound types and the park’s restorative effects and deficiencies for users. We divided the park into eight areas in Xiangjiang Park in June 2022 and used the questionnaire to conduct a field sampling survey to investigate park users at a total of 38 resting facilities in each area, and the distribution of the areas and resting facilities are shown in Figure 2.

The respondents completed the questionnaire under informed conditions, and a total of 220 questionnaires were distributed during the survey, of which 52.27% were completed by males, 13.57% were of the participants were under 18 years old, 32.94% were 19–34 years old, 32.68% were 35–54 years old, and 20.89% were over 55 years old or older. Those who chose more than 70% of uniform answers consecutively in the questionnaire were regarded as invalid samples, and after excluding the invalid samples, the total number of valid samples was 204, with an efficiency rate of 92.73%. The basic information of the survey in each region is shown in Table 1.

As an important factor in a restorative environment, sound types are also very closely related to the working life of urban residents. As a major public activity place for urban residents to relieve stress and relax, the actual perception of different sound types in urban parks is particularly important, and the impact of different sound types on users’ physical and mental health should also be the focus of the research on the restorative environment of the acoustic environment. SPSS software was used to analyze the results of the park participants’ perception of the park sound types in the questionnaire, and the main types of sound environments that can be easily perceived in this park are summarized in Table 2.

During the survey, visitors who stayed in each activity area of the park for more than 5 min were recorded and counted. The analysis was performed with Arc GIS Pro 2.8 to obtain the more popular rest–activity locations in the park, and their activity distribution is shown in Figure 3.

### 2.2. Current Situation and Selection of Measuring Points

In the current status survey of the park, we used field measurement to obtain current the status data. The park was divided into several square grids of equal size (20 m × 20 m), one measurement point was placed in the center of each grid, and an AWA5680 sound level meter with a measurement range of 30–130 dB (A) was used, according to the “acoustic environmental noise measurement method” (GB/T 3222-94). The current sound pressure level of the park was collected from 8:00–10:00 a.m. in the morning and between 14:00–16:00 p.m., and the height of the measuring point was 1.5 m from the ground. The sound pressure level research measurement point data information is shown in Table 3.

After the measurement, the data were input into Origin 2021 for data visualization as shown in Figure 4.

The buildings around this park are mainly residential houses, cultural and educational institutions, and administrative offices, which belong to the Class 1 sound environment functional area according to the sound environment quality standard GB3096-2008, with a daytime ambient noise limit value of 55 dB (A). According to the actual situation, a total of seven representative pilot sites were selected for the study, combined with the distribution map of residents’ activities and the sound environment distribution map. The environment of each test site is shown in Figure 5.

### 2.3. Physiological Test Methods and Procedures

Today’s urban residents are prone to anxiety, depression, and other adverse emotions under the pressures of work and life, and such negative emotions can increase vascular resistance, raise blood pressure [25,26], and alter normal heart rate fluctuations [27]. At the same time, the rapid development of urbanization has led to an increasingly complex sound environment, and environmental noise can also affect blood pressure and heart rate, thus affecting human health [28]. Blood pressure (BP) and electrocardiogram (ECG) tests are now widely used as important methods to determine stress health [29].

ECG is the phenomenon of changes in the bioelectrical signal of the heart during the activity cycle, which is usually displayed by an electrocardiogram (ECG), and in the time domain, this mainly includes the time domain parameters of heart rate (HR), R-wave amplitude, and heart rate variability (HRV). HR is calculated as the number of fluctuations in the cardiac cycle in a minute, which varies with physiological and environmental changes, and is usually influenced by negative emotions such as stress, reflecting the perceived state of stress. HRV is the variation in the differences between heartbeat cycles one by one and can reflect the working condition of autonomic nerves. In addition, HRV is a non-invasive and convenient indicator for studying autonomic activity, and the main parameters are the percentage of the number of adjacent sinus intervals with differences greater than 50 ms in the total number of sinus beats (PNN50), the standard deviation of NN intervals (SDNN), the standard deviation of NN intervals (SDANN), the root mean square standard deviation (RMSSD), etc. Studies have shown that HRV indicators can assess threat, safety, and stress in the body; higher HRV is associated with parasympathetic innervation within the autonomic nervous system, which indicates that the body is in relaxation or a digestive mode; on the other hand, lower HRV is usually associated with elevated sympathetic activity within the autonomic nervous system, which is an indication of elevated stress or a stress response [30].

There are long-range (12–24 h) and short-range tests (t ≤ 5 min) for obtaining HRV by ECG testing, and SDANN and RMSSD are commonly used in short-term HRV analysis [31]. In the short-range measurements of HRV, the RMSSD was generally 34.48 ± 22.10 (ms) in young men aged 20–29 years and 39.29 ± 23.49 (ms) in women. Therefore, the root mean square of the difference between the heart rate (HR) and NN interval (RMSSD) was chosen as an index to assess changes in mental stress in this study. The measurement instrument was a TE-5100Y-C portable ambulatory ECG recorder for real-time monitoring.

For the selection of participants for the physiological index test, we selected volunteers from undergraduate and graduate students who volunteered to participate in the experiment and were informed that they would be taking a mid-term exam. However, each individual may not have the same stress response, so we assessed their psychological stress status by completing the perceived stress scale (PSS) to ensure that the participants were at the same stress level. We also recorded the real-time physiological indicators (HR and RMSSD) of each volunteer, and finally, we selected volunteers with a score of 34 or more on the perceived stress scale as experiment subjects.

The perception of stress scale (PSS) is an individual’s psychological response, subjectively expressed as physical and mental tension and discomfort. PSS [32] is an internationally used scale for measuring psychological stress, which has been validated and promoted in related studies in other countries and has high reliability and validity [33].

A total of 22 subjects screened on the PSS-14 scale were recruited for this study, and were asked to avoid coffee, alcohol, and strenuous exercise starting 24 h before testing, not to smoke, and to get at least 8 h of sleep starting 8 h before testing. In addition, the participants were asked to wear an eye patch during the test to reduce errors in physiological responses due to visual factors.

Professor Tingzhong Yang made the necessary modifications to the structure and entries of the English version of the PSS, according to the Chinese cultural background, resulting in a Chinese version of the stress perception scale suitable for use in China, as shown in Table 4.

In the process of conducting stress the recovery test in the acoustic environment of urban parks, the stress recovery of a stressed population after visiting a space in an urban park is a short-term transient change. Other related scales, such as the depression anxiety stress scale (DASS-21) [34], the symptom check list (SCL-90) [35], and the psychosomatic tension relaxation inventory (PSTRI) [36], which assess the state of physical and mental stress over time, are not fully applicable to this study in terms of temporal validity. Domestic and international studies have shown that positive emotions can undo and restore the activation state of various cardiovascular activities caused by stress and bring them back to normal levels quickly.

Therefore, the positive and negative affect scale (PANAS) [37] was used as a measure of stress recovery in the testing phase of the acoustic environment and physiological change study. The PANAS is a state mood scale, which is mainly influenced by situational factors, has been widely used in the assessment of mental health status at home and abroad, and has shown good performance. The Chinese version of PANAS was tested by Professor Yang Tingzhong and is suitable for use in the Chinese population.

A total of 22 subjects screened by the PSS-14 scale were recruited for this study, with a male to female ratio of 1.2:1 and a mean age of 24 years. The participants were asked to avoid coffee, alcohol, and strenuous exercise 24 h before the test, refrain from smoking and get at least 8 h of sleep 8 h before the test. The subjects were accompanied by a recording officer at each experimental site and were assisted by the recording officer to put on the ECG equipment and the blindfold, which was used to reduce errors in physiological response due to visual factors. After this, the experiment began and the subjects were asked to listen to the type and variation of sound types at the experimental sites for 5 min at each site and record the ECG data of the subjects. The process of filling in the sound factor type questionnaire in the interval after the end of each measurement is one experimental unit, and the above steps are repeated for the next park sample experiment. The flow of the experimental unit is shown in Figure 6.

## 3. Results and Analysis

### 3.1. Analysis Method

Variance, correlation tests, and multiple regression analyses were performed with SPSS software for the sound types and experimental samples in acoustic environments.

Firstly, the normality of the collected data was analyzed using the Kolmogorov–Smirnov test, and the results showed that the quantitative data of the percentage of sound environment types such as birdsong, leaf sound, water flow, wind sound, conversation sound, playful sound, music sound, sports sound, car sound, horn sound, and radio sound were all statistically significant with *p*-values greater than 0.05. Secondly, the Pearson correlation coefficient was used to initially clarify the influence relationship between each sound type and the experimental samples. Finally, multiple linear regression analysis was used to construct a regression model between the types of acoustic environment and physiological changes, and the research process is shown in Figure 7.

### 3.2. Analysis Results

Descriptive statistics of the analysis data are shown in Table 5. Pearson correlation coefficients between the urban park sound environment and HR and RMSSD are shown in Table 6, Table 7 and Table 8.

Pearson’s correlation analysis showed that there was a correlation between HR and 11 sound categories: bird song, leaf sound, water flow, conversation, play, music, movement, car sound, horn sound, radio sound, and mechanical sound. There was a significant correlation between HR and bird song, leaf sound, water flow, conversation, music and radio sound, while there was a significant positive correlation between HR and playful sound, sports sound, car sound, horn sound, and mechanical sounds. The effect of conversation sound (0.05 > *p* = 0.024 > 0.01) on HR was weaker than the other sound types, while wind sound (*p* = 0.147 > 0.05) and footsteps (*p* = 0.569 > 0.05) had no significant correlation with HR.

For the RMSSD, it showed a significant positive correlation with bird song, leaf sound, water flow, and wind sound, and showed a significant negative correlation with conversation sound, playful sound, sports sound, car sound, horn sound, and machine sound, and the effects of wind sound (0.05 > *p* = 0.037 > 0.01) and playful sound (0.05 > *p* = 0.023 > 0.01) on RMSSD were weaker than other sound types, while footsteps, music, and radio sound did not significantly affect the RMSSD with *p* values (*p* = 0.119, 0.587, and 0.298).

Overall, the effects of bird song, leaf sound, car sound, and whistles were dominant in the HR and RMSSD. This result tentatively suggests that bird song and leaf sound have a more significant positive effect on mental stress relief, which is consistent with the findings of Eleanor Ratcliffe et al. [38]. In contrast, car sounds and whistles played a limited role in the recovery from mental stress.

In addition, the regression analysis was able to show more specifically the trend of the influence of different acoustic environment types on the physiological indicators. Since none of the footsteps passed the correlation test of the HR and HRV indicators, they were only used as a reference in the regression analysis. The descriptive statistics of the data from the multiple linear regression analysis are shown in Table 9 and the models of HR and RMSSD influenced by the type of acoustic environment are shown in Table 10 and Table 11.

It can be seen from the table that each type of acoustic environment has an effect on both indicators of HRV for assessing changes in mental stress (R^2^ > 60); both models passed the F test with test coefficients of 64.947 and 71.425, respectively, and the variance inflation factor (VIF < 5) also makes the multiple linear regression model more reliable.

Overall, the correlation between wind and radio sound and HR and RMSSD in the park sound environment was not significant and the Beta value was small, which suggests that the effect of wind and radio sound changes on the mental stress recovery of the testers was weak. In addition to this, the effects of other types of acoustic environments on HR and RMSSD were significant, but some significant differences still existed between them.

## 4. Conclusions and Recommendations

### 4.1. Conclusions

In this study, we found that different types of sound environments have different effects on human physiological indicators by combining sound environments with physiological indicators. Through correlation analysis and multiple linear regression analysis, we clarified the influencing relationships between ECG indexes and different types of sound, and the following conclusions were drawn from the research mentioned above:(1)The distribution of the type of sound environment in the city park is influenced by the external factors of the park. The average sound pressure level at the periphery of the park is 67.21 dB (A), and the main influencing factor of this phenomenon is traffic sound, which causes the sound pressure level of the sound environment in this part of the area to exceed the limit of 55 dB (A) specified in GB3096-2008.(2)Among the effects of different sound types on physiological changes in urban parks, significant correlations existed for all sound types except for the weak correlations for wind and radio sounds. Natural sounds have the strongest positive effect on physiological indicators and are more helpful in relieving mental stress. When the proportion of natural sounds reached 40.4% to 45% of the whole park sound environment, the average heart rate was 68.3 beats/min and the RMSSD was 47.89 ms; mechanical sounds had the strongest negative effect on physiological indicators and limited the recovery of mental stress. When the proportion of mechanical sound reached 35.2–38.5%, the average heart rate was 78.16 beats/min and the RMSSD was 27.59 ms; human activity sound was at an intermediate level. When the proportion of human activity sound reached 42–45.5%, the HR was 74.68 beats/min and the RMSSD was 29.32 ms. It is easy to see that natural sound is a massive help in mental stress recovery.(3)This study finally attempted to construct a multiple linear regression model with the type of acoustic environment and ECG indicators as variables, which makes it possible to reasonably design the proportion of sound types in the acoustic environment according to physiological needs, indicating that we are theoretically able to design the corresponding acoustic environment by setting the physiological indicators as the expected values for environment creation, which has some practical significance for the construction of restorative environments. Still, the method is only an attempt in this study and needs to be confirmed by further research.

By combining sound types with physiological indicators in urban parks, in terms of theory, this study extends the applicability of stress recovery theory and attention recovery theory to a certain extent that is no longer limited to the landscape environment only. In terms of empirical evidence, this study confirms that sound type is also an important influence on the restorative environment from an objective perspective by quantifying the percentage of park sound types through physiological experiments. However, this study still has shortcomings. Quantifying the types of park sound environments in the form of subjective questionnaires makes this study subjective. Therefore, in the future, we need to explore an objective method to quantify the complex types of sound in public spaces. In addition, because the standard values of physiological indicators vary from person to person and the sample size is relatively small, we were not able to explore exactly when the proportion of different types of sound reaches a certain level to achieve the best restorative benefits of the restorative environment, which we believe should be the main direction for future research on the restorative benefits of the sound environment at an objective level.

### 4.2. Optimization Suggestions

Along with the in-depth research on the effect of urban park acoustic environments on health recovery, based on the conclusions of this study and combined with the previous research results, the following suggestions are made for the future planning and design of urban parks with a holistic to local design idea.

As a whole, the future planning and design of urban parks should not only carry out dynamic and static partitioning in terms of useful functions, but also clarify the functions of the acoustic environment in different spaces and control the dynamic and static rhythm of the acoustic environment to meet the needs of different users. Urban parks are not only a tool to absorb urban noise and adjust the local environment [39], but they are also a place for people’s leisure activities. While using urban parks as a tool to fight against external adversity, it is necessary to fully consider the restoration benefits of parks.

Locally, the first step is to reduce the impact of traffic noise on urban parks. The sound of moving cars and honking not only affects the health of residents, but also has a disturbing effect on vocal organisms such as birds [40,41], thus affecting the content of natural sounds such as birdsong in the park, which in turn weakens the health restoration effect of urban parks, and the impact of traffic noise can be mitigated to some extent by matching shrubs and trees in the green belt on both sides of the road and using plants as a barrier [42]. Second, more natural sound should be introduced. The flow of water can produce sound and people like this sound second only to birdsong. The main bodies of flowing water are fountains and waterfalls. The sound of flowing water can effectively mask other unwanted sounds [43,44]. Birdsong, whirring, and leaf sounds are not only psychologically acceptable to park users, but also physiologically contribute to the recovery of human health. By adjusting the types of plant communities in a park, biodiversity can be effectively increased and natural sounds can be enriched. Finally, to improve the more complex and chaotic sound environment, a sound environment landscape can be created to divert people’s attention from negative sound or by artificially playing soothing music to mask the noisy surrounding sound environment and create a local activity atmosphere. In addition, not only should there be sufficient consideration in the sound environment design but also appropriate improvement in park management, maintenance of the overall park environment and landscape facilities, and a reduction in behavioral interference with the park sound environment.

## 5. Discussion

In this study, we further explored the influence mechanism of sound types on physiological health through the combination of sound environment types and electrocardiogram physiological indicators in urban parks and proposed the optimization method of sound environment quality in urban parks according to the factor of sound types. In addition, the use of portable ECG recording equipment avoids invasive effects on the subjects and reduces the psychological burden on the test subjects. At the same time, the popularity of portable devices also makes the sites for human–computer environment studies more flexible and versatile. Using the actual site as an experiment, compared to watching videos and listening to sounds in an indoor lab, improves the accuracy of environmental and physiological studies. However, at the same time, the outdoor environment is more complicated. Some uncontrolled or overlooked variables may influence the results of the experiment. This requires continuous improvement of experimental methods and quantitative methods of environmental variables for urban parks.

In terms of stress state indicators, unlike previous studies that used only subjective questionnaire scales, the introduction of real-time objective physiological indicators, HR and RMSSD, as evaluation parameters avoids the shortcomings of overly subjective studies, and the combination of subjectivity and objectivity makes the results more convincing. Although no studies have shown that HR and RMSSD values in a certain range are considered stress-relieving states—some are just the normal range of values for the index—some studies have shown that stress states are positively correlated with heart rate and negatively correlated with heart rate variability, which is consistent with Table 10 and Table 11. The finding that natural sound has a better restorative effect on mental stress is consistent with and extended the theory of Kaplan and Ulrich. From an objective point of view, it is not just natural landscapes that can relieve stressful situations, natural sounds also have the same benefits. At the same time, our study is consistent with some of the conclusions of Zhao [45], and Ratcliffe [38,46], who used semi-structured interviews to analyze the effects of bird sounds on stress recovery. Another part of their conclusion was that not all birdsong has a stronger restorative effect, the reduced recovery of birdsong may be related to low levels of an existing connection to nature. Thus, with the deepening of cross-disciplinary research, more precise quantitative analysis methods and studies are needed for the impact of the public acoustic environment on people’s health in the future.

The main limitation of this study is that the object of the study is relatively single and only one park was studied, so the conclusions may have certain regional limitations. This requires researchers from different regions and countries to jointly explore and study the distribution rules and influencing factors of sound types in different types of parks. The main contribution of this study is to provide a more objective and accurate method to explore the effect of the type of urban park sound environment on mental stress recovery and from a more subjective point of view, the influence mechanism of sound types on participants in an urban park environment was analyzed.

However, there are still shortcomings in this study. The acoustic environment involves more than just the type of sound; environmental temperature [47] and biological diversity [48] may be factors affecting the sound environment of urban parks, which may need to be studied from an ecological perspective. In addition to this, the testers selected for this study are all school students with an average age of 24 years, and the relationship between changes in physiological responses to perceiving the sound environment varies among people of different age groups, which may affect the relationship between sound type and mental stress. In the future, we may be able to carry out a targeted and more detailed restorative design for different service groups in the sound environment of urban parks by comparing the different reaction mechanisms of different age groups. There is also the influence of seasonal factors, and this study was chosen to be conducted in the same season to avoid the influence of seasonal changes on the sound environment. In the future, we can also build on this study to explore the impact of seasonal changes in the acoustic environment on recovery benefits. The factors listed in this discussion are not the only factors affecting the health of the acoustic environment. Therefore, further research is needed to analyze these influencing factors.

## Figures and Tables

**Figure 1 ijerph-20-01082-f001:**
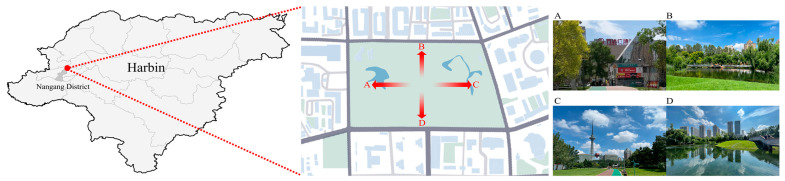
Park location and surrounding environment. (**A**) Park south side environment. (**B**) Park west side environment. (**C**) Park north side environment. (**D**) Park east side environment.

**Figure 2 ijerph-20-01082-f002:**
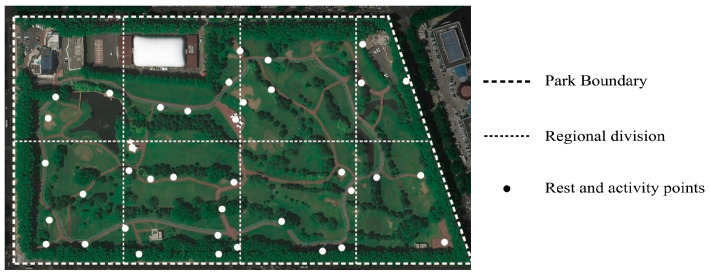
Regional division and rest facilities distribution map.

**Figure 3 ijerph-20-01082-f003:**
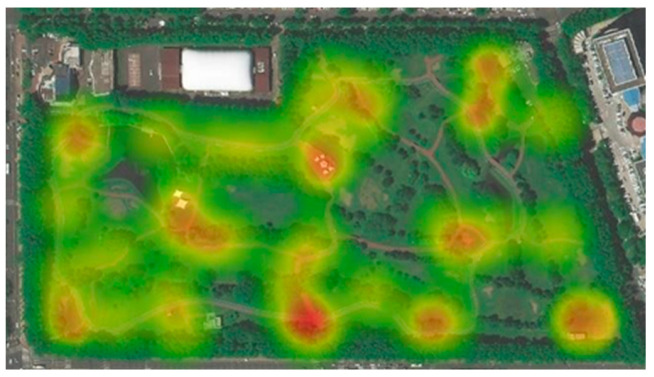
Residential activity distribution map.

**Figure 4 ijerph-20-01082-f004:**
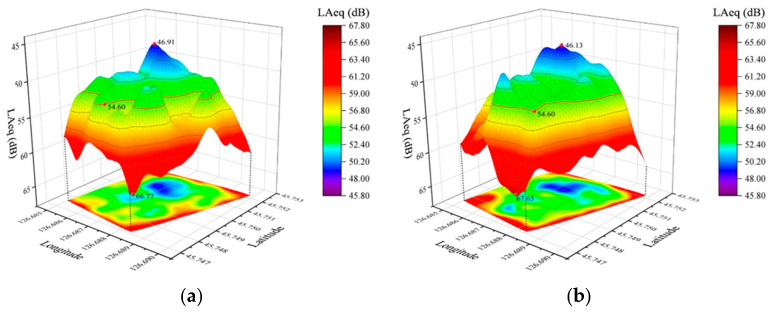
Three-dimensional map of sound pressure level distribution. (**a**) 8:00–10:00 a.m., and (**b**) 14:00–16:00 p.m.

**Figure 5 ijerph-20-01082-f005:**
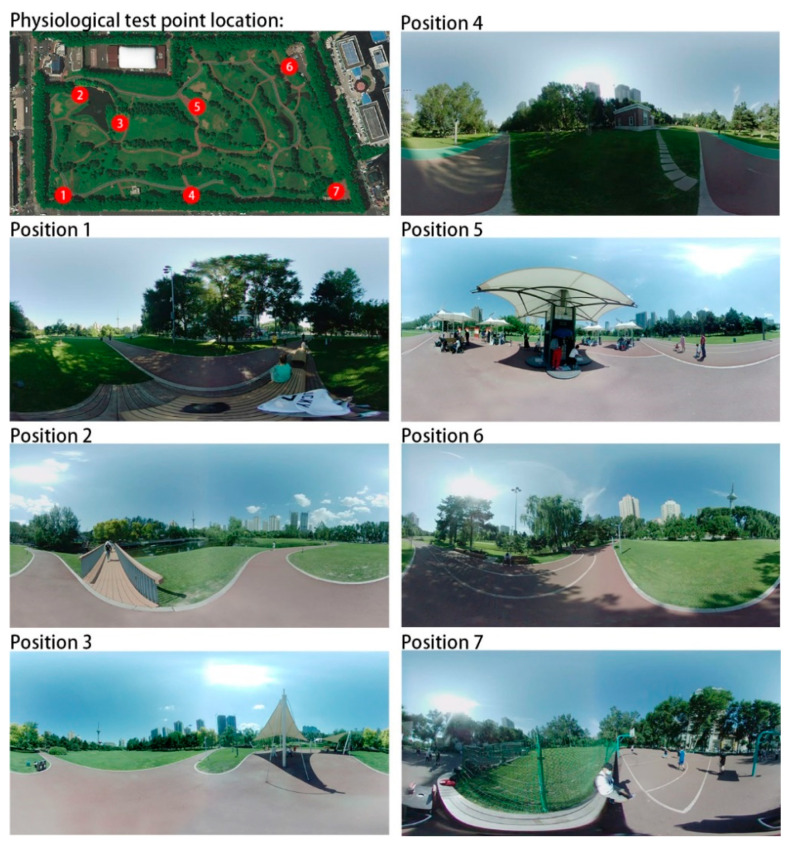
Physiological test point location and environment.

**Figure 6 ijerph-20-01082-f006:**
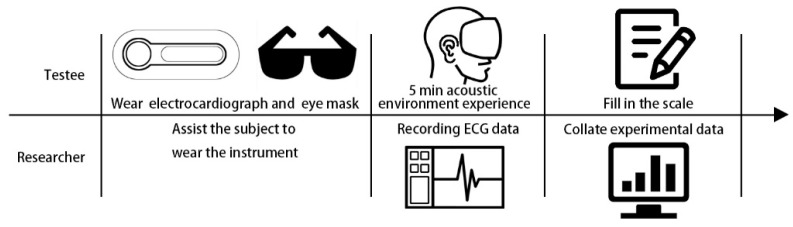
Flow chart of experimental unit.

**Figure 7 ijerph-20-01082-f007:**
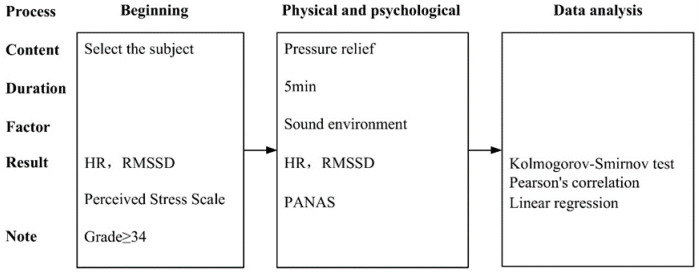
Experimental research process.

**Table 1 ijerph-20-01082-t001:** Basic survey information table for each region.

	Area Number							
	A	B	C	D	E	F	G	H
Number of rest and points	3	4	4	3	5	11	5	3
Number of questionnaires issued	27	28	28	27	25	33	25	27

**Table 2 ijerph-20-01082-t002:** Main types of acoustic environment in the park.

Sound Source Properties	Sound Environment
Natural Sound	Birdsong, leaf sound, purl, and wind
Social Sound	Conversation, playful sound, footsteps, music sound, and sports sound
Mechanical Sound	Car sound, whistles, radio sound, and machine sound

**Table 3 ijerph-20-01082-t003:** Sound pressure level research measurement point data description statistics.

No.	8:00–10:00			14:00–16:00		
	Max	Min	Mean	Max	Min	Mean
A	67.2	57.4	61.73	65.2	58.7	62.35
B	65.1	50.9	57.33	67.6	53.1	56.84
C	62.1	45.9	56.24	65.4	49.7	55.4
D	66	50.4	55.40	68.8	50.2	55.16
E	60.4	48.9	54.35	67.9	49.7	54.81
F	61.1	49.4	54.61	61	50	53.57
G	64	47.1	52.13	58.4	45.9	52.05
H	61.3	49.5	54.08	61.3	49.2	54.26
I	67.8	46.7	53.57	63.9	47.4	53.38
J	64.1	50.2	55.17	67.5	50.9	57.87
K	61	50.8	55.88	63.9	51.7	57.07
L	61.5	57.4	58.99	62.3	54.9	57.77

**Table 4 ijerph-20-01082-t004:** Perceived Stress Scale, PSS.

Question	Never	Almost Never	Sometimes	Fairly Often	Very Often
1. How often have you been upset because of something that happened unexpectedly?	0	1	2	3	4
2. How often have you felt that you were unable to control the important things in your life?	0	1	2	3	4
3. How often have you felt nervous and stressed?	0	1	2	3	4
4. How often have you dealt successfully with irritating life hassles?	0	1	2	3	4
5. How often have you felt that you were effectively coping with important changes that were occurring in your life?	0	1	2	3	4
6. How often have you felt confident about your ability to handle your personal problems?	0	1	2	3	4
7. How often have you felt that things were going your way?	0	1	2	3	4
8. How often have you found that you could not cope with all the things that you had to do?	0	1	2	3	4
9. How often have you been able to control irritations in your life?	0	1	2	3	4
10. How often have you felt that you were on top of things?	0	1	2	3	4
11. How often have you been angered because of things that happened that were outside of your control?	0	1	2	3	4
12. How often have you found yourself thinking about things that you have to accomplish?	0	1	2	3	4
13. How often have you been able to control the way you spend your time?	0	1	2	3	4
14. How often have you felt difficulties were piling up so high that you could not overcome them?	0	1	2	3	4

**Table 5 ijerph-20-01082-t005:** Descriptive statistics of variable data.

Variable Type	Variable Name	Min	Max	Mean
Independent variable	Birdsong	2.30	18.49	7.99
	Leaf sound	2.85	13.92	8.33
	Purl	3.55	14.21	7.94
	Wind	2.72	12.61	8.37
	Conversation	3.30	16.07	7.25
	Playful sound	3.38	12.34	8.30
	Footstep	1.57	13.04	8.21
	Music sound	3.30	14.41	7.44
	Sports sound	2.14	15.18	6.79
	Car Sound	3.61	15.67	7.78
	Whistles	3.54	12.94	7.13
	Radio sound	2.38	14.98	7.41
	Machine sound	1.94	11.91	7.14
Dependent Variable	HR	61.00	87	73.72
	RMSSD	15.00	59	32.85

**Table 6 ijerph-20-01082-t006:** Pearson correlation between natural park sounds and heart rate variability.

	Natural Sound			
	Birdsong	Leaf Sound	Purl	Wind
HR	−0.355 **	−0.278 **	−0.207 **	−0.068
RMSSD	0.425 **	0.233 **	0.219 **	0.097 *

** at the 0.01 level (two-tailed), the correlation is significant; * at the 0.05 level (two-tailed), the correlation is significant.

**Table 7 ijerph-20-01082-t007:** Pearson correlation between human activity sound and heart rate variability in the park.

	Human Activity Sound				
	Conversation	Playful Sound	Footsteps	Music Sound	Sports Sound
HR	0.105 *	0.166 **	0.027	−0.141 **	0.294 **
RMSSD	−0.185 **	−0.106 *	0.073	−0.025	−0.427 **

** at the 0.01 level (two-tailed), the correlation is significant; * at the 0.05 level (two-tailed), the correlation is significant.

**Table 8 ijerph-20-01082-t008:** Pearson correlation between park mechanical sound and heart rate variability.

	Mechanical Sound			
	Car Sound	Horn Sound	Radio Sound	Machine Sound
HR	0.390 **	0.356 **	−0.134 **	0.123 **
RMSSD	−0.358 **	−0.154 **	0.049	−0.127 **

** at the 0.01 level (two-tailed), the correlation is significant.

**Table 9 ijerph-20-01082-t009:** Data descriptive statistics for multiple linear regression analysis.

Tier 1 Indicator	Variable Name	Variable	Multiple (%)
Natural Sound	Birdsong	X_1_	7.989
	Leaf Sound	X_2_	8.326
	Water Flow	X_3_	7.937
	Wind	X_4_	8.367
Human activity sound	Conversation	X_5_	7.255
	Playful Sound	X_6_	8.302
	Music Sound	X_7_	7.444
	Sports Sound	X_8_	6.795
Mechanical sound	Car Sound	X_9_	7.786
	Horn Sound	X_10_	7.139
	Radio Sound	X_11_	7.406
	Machine Sound	X_12_	7.140
Explained variables	HR	Y_1_	73.723
	RMSSD	Y_2_	32.474

**Table 10 ijerph-20-01082-t010:** Linear regression model of acoustic environment type and HR.

	Unstandardized		Standardized Coefficients	VIF	*p*	R^2^	F
	B	Std. Error	Beta				
Constant	77.801	4.762	-	-	0.000 **	0.634 0.624	F = 64.947 *p* = 0.000 **
Birdsong	−0.446	0.074	−0.360	4.382	0.000 **		
Leaf sound	−0.555	0.072	−0.386	3.093	0.000 **		
Water Flow	−0.605	0.078	−0.448	4.067	0.000 **		
Wind	0.062	0.065	0.041	2.327	0.341		
Conversation	−0.199	0.085	−0.125	3.530	0.020 *		
Playful sound	0.317	0.084	0.204	3.589	0.000 **		
Music sound	−0.071	0.096	−0.042	4.058	0.461		
Sports sound	0.482	0.056	0.305	1.563	0.000 **		
Car sound	0.407	0.080	0.252	3.016	0.000 **		
Horn sound	0.496	0.086	0.271	2.703	0.000 **		
Radio sound	−0.115	0.060	−0.080	2.177	0.058		
Machine sound	−0.199	0.100	−0.102	3.227	0.048 *		

** and * represent 1% and 5% significance levels, respectively.

**Table 11 ijerph-20-01082-t011:** Linear regression model between acoustic environment type and RMSSD.

	Unstandardized		Standardized Coefficients	VIF	*p*	R^2^	F
	B	Std. Error	Beta				
Constant	45.437	8.393	-	-	0.000 **	0.656 0.647	F = 71.425 *p* = 0.000 **
Birdsong	0.521	0.130	0.232	4.382	0.000 **		
Leaf sound	0.839	0.127	0.321	3.093	0.000 **		
Water flow	0.991	0.137	0.404	4.067	0.000 **		
Wind	0.141	0.114	0.052	2.327	0.218		
Conversation	−0.094	0.151	−0.032	3.530	0.534		
Playful sound	−0.676	0.148	−0.239	3.589	0.000 **		
Music sound	−0.585	0.169	−0.193	4.058	0.001 **		
Sports sound	−1.526	0.100	−0.530	1.563	0.000 **		
Car sound	−0.857	0.141	−0.292	3.016	0.000 **		
Horn sound	−0.411	0.152	−0.123	2.703	0.007 *		
Radio sound	−0.134	0.106	−0.051	2.177	0.208		
Machine sound	−0.164	0.176	−0.046	3.227	0.354		

** and * represent 1% and 5% significance levels, respectively.

## Data Availability

The original contributions presented in the study are included in the article; further inquiries can be directed to the corresponding authors.

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
