# Peer review of "Effects of Acoustic Environment Types on Stress Relief in Urban Parks"

_ijerph, 2023, doi:10.3390/ijerph20021082_

Round 1
Reviewer 1 Report
Dear author(s),
Thank you for the opportunity to review this paper. I agree that this is an important and pertinent topic. There are a few areas where I would encourage the authors to give further thought, as follows:
· Novelty and originality of the research must be added to the abstract.
· The introduction should clearly illustrate (1) what we know (the key theoretical perspectives and empirical findings) and what do we not know (major, unaddressed puzzle, controversy, or paradox does the study addresses, or why it needs to be addressed and why this matters). And, (2) what will we learn from the study and how does the study fundamentally change, challenge, or advance scholars’ understanding. Much sharper problematization is required so that the introduction draws the reader into the paper. The introduction therefore needs to do a better job in setting the stage for the articulation of the theoretical contributions of the study. At the end of the introduction, we should have a clear idea of what the paper is about (i.e. its motivation, the gap in understanding that the paper is trying to address and summary of theoretical contributions).With references of 2022- 2020-2021.
Paragraph 1, with no references, explaining the context of the research.
Paragraph 2, with references, explaining very generally what we know about the topic introduced in Paragraph 1.
Paragraph 3 explaining what we need to find out.
Paragraph 4 explaining briefly what this paper will do to find out, method etc.
Paragraph 5, with no references, explaining the structure of this paper.
· You need a discussion section. The discussion challenges your findings and determines the degree of compatibility with previous research.
· The discussion section needs to highlight what is new in your findings and what we can learn from a study conducted in this interesting and understudied context. Whilst the introduction sets the stage for the study by justifying the relevance of the study, the discussion is the most important section as it is in the discussion that it is all brought together, and the authors illustrates how and why the study findings advance the literature. Therefore, the discussion needs to illustrate the new insights—the contributions—in a clear and compelling manner. In other words, illustrate what we know now that we did not know before or, in effect, to clearly illustrate the contribution of the study to the different bodies of literature. Furthermore, what are the future research directions based on this new framework?
· The authors need to draw substantive conclusions from their results, and suggest, develop recommendations for further research.
· What are the limitations of this research and how can it be solved by other researchers?
Best of luck with the further development of the paper.
Author Response
请参阅附件。

Reviewer 2 Report
The results are interesting and may be useful for urban planning. However, the methods are not clear. The authors should modify the section and add more explanation.
1) Questionnaire for the park users: The questionnaire survey was administered to the park users, general people. The number of respondents and their demographic variables should be presented. Where in the park were the respondents interviewed? How many respondents per site? These may be summarized in a table. The questionnaire may be presented as Appendix.
2) Psychological test and physiological measurement: The subjects of the test and measurement were all students. The author should indicate the distribution of sex and age at least. The experiment procedure is not clear. Where in the park and at how many sites were the experiment conducted? How did the subjects move to the survey sites in the park? Or did the subjects participate in the test/measurement at only one site? How many subjects participated in the experiment at sites? I think that these are important to interpret and evaluate the results.
3) How did the authors associate results from the questionnaire survey to the general people with those from the experiments using the students? These groups are quite different in psychological and physiological aspects. The author might conduct the test/measurement to some general volunteers and compare the results with those of students.
4) There are many abbreviations in lines 144, 145, 175, etc. The authors should show the full spells when they are first presented.
5) Others
Table 3: Add "**" to "0.425"
Line 100: chouse?
lines 219-220: "positive" and "negative" may be opposite from a statistical view point.
line 231: hooting-->footing?
Author Response
请参阅附件。

Round 2
Reviewer 1 Report
Dear author(s),
Thank you for the opportunity to review this paper.
• You need a conclusion section. What are the theoretical and practical implications of your study and which limitations and possible future research emerge from it?
• Theoretical Contributions: Addressing all the points mentioned above will lead to a more in-depth presentation of your data which has a clearer theoretical contribution. What is the theoretical contributions?
• The authors need to draw substantive conclusions from their results, and suggest, develop recommendations for further research.
Reviewer 2 Report
Comments to the authors
I appreciate the authors for their honest revisions to my comments. However, the authors should still revise the following points.
1) Line 175: The full spells of SDNN, SDANN, and RMSSD should be presented.
2) Line 114-115: Too fine numbers are not necessary and the total is not 100%. Please check.
3) If Tables 7 and 8 are the results of the multiple linear regression analysis, add “multiple” to the cation.
What I concern most is the English sentences in the revised manuscript. Since I am not an English native, I cannot correct them. Many sentences are very strange and difficult to read. The whole text should be thoroughly proof-edited by English professionals. For example, the following sentences may be grammatically wrong: lines 52-55, 75-76, 82-87, 104-106, 139-142, 188, 222, 231, 232-235, 243, 254-255, 279-280, 283, 295, 305-307, 338-345, 367, 368-370, 377-378, 380-381, 398-402, and many others. Understandable text is the requisite for the publication.
